# Spring Wheat–Summer Maize Annual Crop System Grain Yield and Nitrogen Utilization Response to Nitrogen Application Rate in the Thermal–Resource–Limited Region of the North China Plain

Meng Liu [1], Zhiqi Ma [1], Qian Liang [1], Yao Zhang [1], Yong'an Yang [2], Haipeng Hou [3], Xidong Wu [1] and Junzhu Ge [1,*]

[1] College of Agronomy, Resources and Environment, Tianjin Agricultural University, Tianjin 300392, China
[2] Tianjin Highquality Agricultural Products Development Demonstration Center, Tianjin 301500, China
[3] Tianjin Agricultural Development Service Center, Tianjin 300061, China
[*] Correspondence: gejunzhu@tjau.edu.cn; Tel.: +86-189-2028-2716

**Abstract:** Spring wheat–summer maize (SWSM) annual crop systems were formed to satisfy the maize grain mechanized harvest thermal requirement in the thermal–resource–limited region of the North China Plain. However, the nitrogen (N) application rate effect on SWSM annual yield formation, N accumulation and utilization were barely evaluated. Two–year field experiments were conducted to evaluate the effects of the N application rate on the annual yield of SWSM, observe N accumulation and utilization, and identify the optimized N application. The experiments were conducted under 5 N levels of 0 (N0), 180 (N180), 240 (N240), 300 (N300), and 360 (N360) kg ha$^{-1}$. The results showed that spring wheat, summer maize and annual cereal yield under the $N_{240}$ and $N_{480}$ treatments obtained the highest grain yield (GY) of 5038, 1282 and 16,320 kg ha$^{-1}$, respectively, and the optimal N application rate was estimated using a linear–plateau model to be 231–307, 222–337 and 463–571 kg ha$^{-1}$ with maximum GY of 4654–5317, 11,727–12,003 and 16,349–16,658 kg ha$^{-1}$, respectively. With the increase in the N application rate, the dry matter accumulation (DM) were significantly increased by 16.9–173.5% for spring wheat and 11.1≈–76.8% for summer maize, respectively; and the annual cereal DM was 15.1–179.7% greater than that with $N_0$ treatment, respectively. Spring wheat, summer maize and the annual cereal total N accumulation (TN) under $N_{360}$ and $N_{720}$ treatments were significantly increased by 5.4–19.1%, 16.6–32.3% and 11.5–26.2%, respectively, compared to the other treatments; however, N use efficiency for biomass and grain production (NUEbms and NUEg) were decreased significantly by 10.9–13.6% and 8.9–20.7%, 6.8–13.8% and 12.2–15.6%, and 5.5–11.7% and 10.0–16.0%, respectively. Meanwhile, the N partial factor productivity (PFPN), N agronomy use efficiency (ANUE), N recovery efficiency (NRE) and N uptake efficiency (NEupk) under the $N_{240}$ treatment for spring wheat and summer maize obtained high levels of 20.99 and 47.01 kg$^{-1}$, 9.27 and 16.35 kg$^{-1}$, 32.53% and 32.44%, and 0.85 and 0.72 kg$^{-1}$, respectively. Correlation analysis showed that the N application rate, TN and NEupk played significantly positive roles on GY, spring wheat spilke grain number, summer maize ear grain number and 1000–grain weight, DM LAImax and SPADmax, while NUEbms, NUEg, PFPN and ANUE always played negative effects. These results demonstrate that spring wheat, summer maize and annual cereal obtained the highest GY being 4654–5317, 11,727–12,003 and 16,349–16,658 kg ha$^{-1}$ with the optimal N application rate 231–307, 222–337 and 463–571 kg ha$^{-1}$, respectively, which provide N application guidance to farmer for spring wheat–summer maize crop systems to achieve annual mechanical harvesting in the thermal–resource–limited region of the North China Plain.

**Keywords:** spring wheat; summer maize; grain yield; yield components; nitrogen application rate; nitrogen accumulation and utilization; thermal–resource–limited region; North China Plain

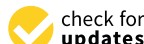

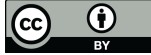

## 1. Introduction

The North China Plain is the main wheat and maize production region in China, producing almost 60–80% of the country's wheat and 35–40% of its maize every year [1]. Nitrogen (N) is one of the most important nutrients for crop growth and yield formation, contributing 30–50% to increases in grain yield [2]. In order to yield increase the cereal grain yield a large amount of N fertilizers being applied in the intensive winter wheat–summer maize double–crop farming system is characterized in China [3–5]. The use of N fertilizer has increased dramatically worldwide [6], and China's agriculture practices account for about 29% of N fertilizer's total global consumption [7].

To date, numerous studies have focused on the positive impacts of N on winter wheat–summer maize yield. Crops require more overall N to produce high grain yield [8]. The yield of wheat grain is increased through high N fertilizer application rates [9], while moderate N fertilizer rates ensure appropriate N accumulation in leaves and N transfer to the spike, as well as increase floret fertility, grain number and yield [10]. However, higher N application rates does not increase crops yield linearly, and excessive N application results in delayed harvesting or plant lodging [11]. In addition, about 30% of the applied N may remain in the soil after harvest in the North China Plain [12,13], causing serious negative environmental impacts such as the eutrophication of surface waters, nitrate leaching, greenhouse gas emissions, soil acidification, etc. [14–18]. In addition, excessive application of N is very common in the intensive winter wheat–summer maize cropping system in the North China Plain, especially in the regions with higher annual yields [19]. The average N application rate for the wheat season is 229–350 kg ha$^{-1}$ and higher than 250 kg ha$^{-1}$ for the maize season based on a national survey of farms [20,21]. This greatly exceeds the supply of N these crops require, leading to considerable N waste and low N use efficiency [22]: only about 30–50% of the applied N is absorbed over the year [23,24], while the wheat–maize crop rotation system's N use efficiency can be as low as 26–31% in the North China Plain [25]. The use of higher N application rates to maintain higher crops yield is unsustainable for intensive winter wheat–summer maize planting systems in the North China Plain, and it is therefore necessary to optimize N application rate to realize crop requirements while reducing N losses as well [26,27]. Numerous studies have focused on rational N management measures to improve annual crop yields and sustainable N utilization [28]. Yin [29] reported that management of the steady–state N balance (SSNB) reduced nitrogen input by 21–28% and either maintained current crop yields or increased them by 6–7%. Moreover, the nutrient expert (NE) system reduced N application by 14.7–29.0% and increased N recovery by 10.8–13.4% [30]. The application of N inhibitors increased N utilization, especially in the winter wheat season when using urease inhibitors, reducing the annual N application rate [31]. In the North China Plain, the average N application rate is 165–211 kg ha$^{-1}$ in winter wheat and 187–250 kg ha$^{-1}$ in summer maize under deficit irrigation, which is the best drip irrigation and fertilization strategy [32]. The economically optimal N rate averages to 150 kg ha$^{-1}$ (130 to 160 kg ha$^{-1}$) for both wheat (6 Mg ha$^{-1}$) and maize (9 Mg ha$^{-1}$) according to the field results from region–wide experiments [20]. Maize even maintained a yield of 6.0–7.5 t ha$^{-1}$ with the N application rate reduced to 120 kg ha$^{-1}$ [33].

Water shortages are an urgent problem for the sustainable development of China's agriculture sector, especially in the North China Plain, which has reached the regional and national limit of underground water extraction [34]. In the winter season, the precipitation is only 174–261 mm, which demonstrates an extreme shortage considering the demand for water in the winter wheat season is above 450 mm [35,36]. Thermal shortage is another limiting factor for winter wheat–summer maize annual crop systems, which can limit annual yields. Thermal utilization has seen synergistic improvement due to traditional planting techniques, such as varieties used, tillage techniques, and sowing and harvest date, etc. [37,38]. These improvements were especially needed in the mechanical harvesting requirements of summer maize grain, which were always restricted to higher grain moisture content due to the long growth period and high thermal demand [39,40]. Several studies



have indicated that delaying the sowing date for winter wheat sowing and the harvest date for summer maize could increase their annual yield significantly without increasing costs [41]. As a result, summer maize's grain moisture content reduced to 14.4–17.3% and its 100–grain weight increased to 22.9–38.4 g (vs. 23.3–37.4 g), thus increasing annual yield by 7.9–6.7%; meanwhile the thermal allocation of summer maize season increased by 497.8–509.6 °C and the thermal utilization rate increased by 5.1–5.6% [41–45].

Former studies have mainly focused on annual winter wheat–summer maize crop systems in the North China Plain. However, identifying the optimal N application rate to achieve higher grain yields, the benefits of N utilization and mechanical grain harvesting for annual spring wheat–summer maize crop systems remain barely understudied, based on the transition from winter wheat to spring wheat to afford summer maize more than 60 days for grain moisture reducing, in the thermal–resource–limited region of the North China Plain. Thus, the main objectives of this study were to: (i) determine the effects of N application rate on the annual yield, dry matter accumulation and crop N accumulation and utilization of spring wheat–summer maize; (ii) analyze the relationship between N accumulation and utilization, cereal grain yield and dry matter accumulation; and (iii) identify the optimized N application rate to provide a theoretical basis for a rational and efficient N fertilizer application rate for spring wheat–summer maize annual crop systems in the thermal–resource–limited region of the North China Plain.

## 2. Materials and Methods

### 2.1. Experimental Sites

We conducted a 2–year annual field experiment, from February to November in 2018 and 2019, at an experimental farm of Tianjin Agricultural University in Dongjituo town, Tianjin, China (117°49′ E, 39°42′ N, 8 m elevation). This farm is located in the thermal–resource–limited region of the North China Plain and has a sub–humid warm temperate monsoon climate, where the traditional annual crop systems are winter wheat–summer maize and spring maize, and spring wheat–summer maize in the last ten years. The properties of the soil, determined from its 0–20 cm–deep layer, are as follows: pH of 8.1, 18.6 g kg$^{-1}$ organic matter, 1.1 g kg$^{-1}$ total N, 64.8 mg kg$^{-1}$ available Olsen P, 296.0 mg kg$^{-1}$ exchangeable K, and 3.1 mg kg$^{-1}$ total soil salt content. The daily rainfall (RF), mean temperature (Tmean), maximum temperature (Tmax) and minimum temperature (Tmin) are shown in Figure 1. The growth degree days for spring wheat were 1907.5 and 1923.6 °C d$^{-1}$ in 2018 and 2019, with total rainfall being 67.0 and 89.4 mm, respectively. During summer maize seasons the growth degree days and total rainfall were 2871.2 and 2880.7 °C d$^{-1}$, and 387.7 and 344.5 mm, respectively.

### 2.2. Experimental Design and Cropping Management

Experiments were conducted using a single–factor randomized block design, where the N application rates for spring maize and summer maize were the same, with 0 (N0), 180 (N180), 240 (N240), 300 (N300) and 360 (N360) kg ha$^{-1}$. Each plot was 7 m long and 4.2 m wide with a 1 m–wide gap between plots, and the experiment was conducted in triplicate. The spring wheat variety was Jinqiang 8. The seeding dates were 25–28 February, and the harvest date was 25 June. The row spacing was 20 cm, with a seeding amount of 450 kg ha$^{-1}$. Nitrogen was applied at the seeding, elongation and heading stages at 40%, 30% and 30%, respectively, while $P_2O_5$ and $K_2O$ were applied at the seeding stage at 90 kg ha$^{-1}$. The field was irrigated with 50 mm each time in the elongation and heading stages based on the farmers management measures. The summer maize variety was Jingnongke 728, which was planted immediately after the spring wheat harvest on 25 June, with the harvest date being 10 November, which was delayed by 35 d compared with the normal harvest date. The row spacing was 60 cm, with plant densities of $9.0 \times 10^4$ plants ha$^{-1}$. Nitrogen was applied at the planting, jointing, and V12 leaf stages at 50%, 30% and 20%, while 120 kg ha$^{-1}$ of $P_2O_5$ was applied at the seeding stage, and 150 kg ha$^{-1}$ of $K_2O$ was applied at the seeding and V12 leaf stages at 50% and 50%. The field irrigation

was 75 mm each time in the jointing and silking stages based on the farmers management measures. Diseases, pests and weeds were controlled by using chemicals throughout the growing seasons.

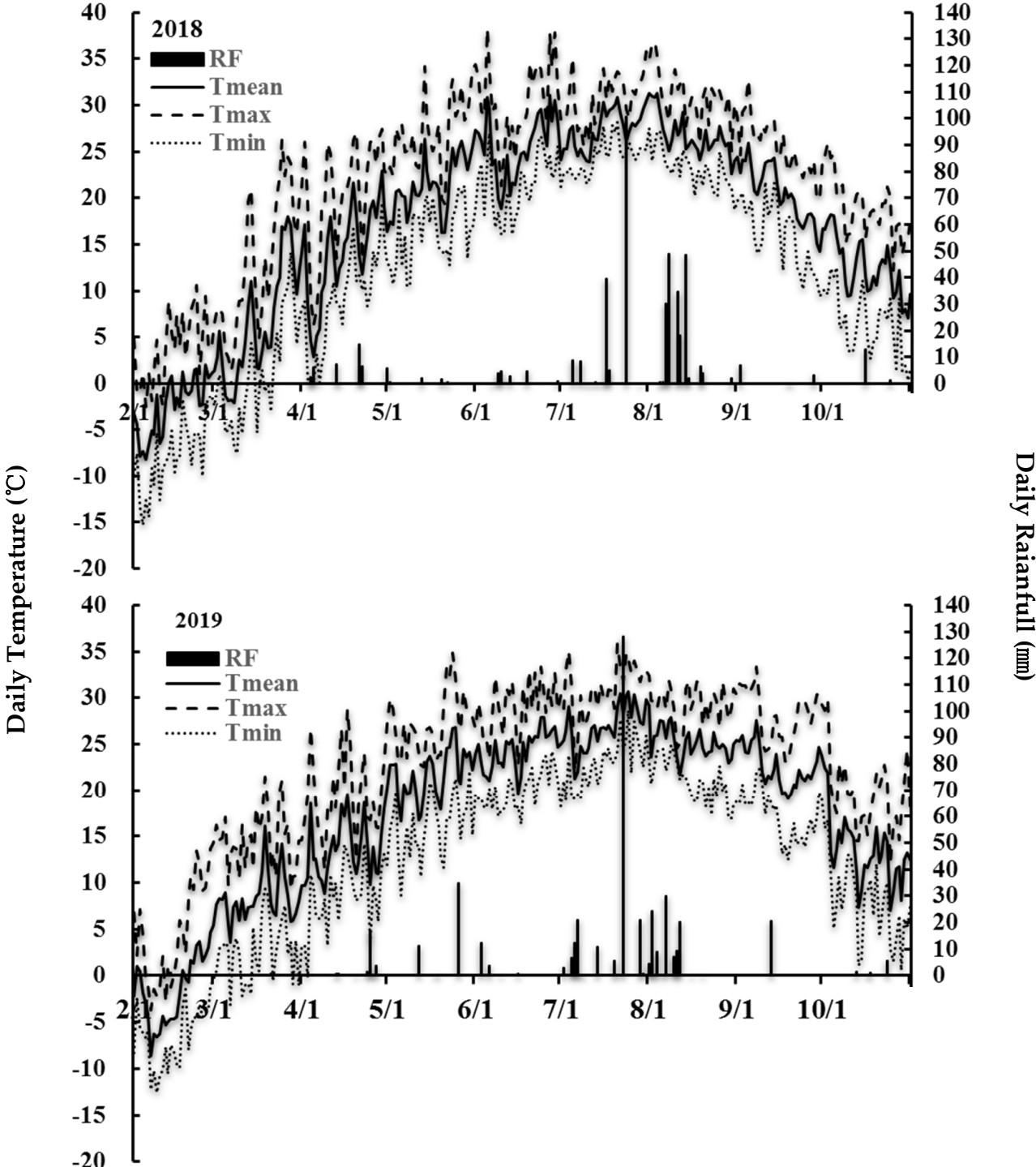

**Figure 1.** Daily meteorological data over the growing duration of spring wheat and summer maize in 2018 and 2019.

*2.3. Measurement Methods*

2.3.1. Grain Yield and Its Components

At the spring wheat harvest stage, all plants covering a 1 × 1 m area from each plot were sampled in triplicate, packed into mesh bags and then dried in the air for 7–10 day.

Then, threshing was conducted, and the dry weight and moisture content were measured. Meanwhile, the 1 m two–row method was employed to estimate the spike number, and 50 representative spikes were randomly sampled from the yield estimation point to estimate yield components, namely, the spike grain number and the 100–grain weight. The spring wheat grain yield and 100–grain weight were calculated at a grain moisture content of 13%. Similarly, at the summer maize harvest stage, 20 ears were collected from the middle two rows in each plot, and the ear grain number and 1000–grain weight were counted. Then, the dry weight and moisture content were measured, and the grain yield and 1000–grain weight were calculated at a moisture content of 14%.

### 2.3.2. Dry Matter Accumulation (DM)

At the spring wheat harvest stage, 100 representative plants were randomly selected from the grain yield sampling point in each plot. The aboveground parts of the plants were divided into stems, leaves, sessile, glumes, and grain and then packed into paper sampling bags. Additionally, at the summer maize harvest stage, 5 representative maize plants with stable growth were randomly selected in each plot. The aboveground parts of the plants were divided into stems, sheaths, leaves, bracts, spike stalks, and grain and then packed into paper sampling bags. All the samples were kilned at 105 °C for 30 min, dried at 75 °C until constant weight and weighed on a balance with an accuracy of 0.01 g. Each plot's plant fraction was passed through a 2 mm sieve for chemical analysis.

### 2.3.3. Maximum Leaf Area Index (LAImax) and SPAD Values (SPADmax)

At the spring wheat heading stage, two 30 cm inner rows of wheat plants from each experimental plot were sampled, all the leaves were removed from the plants, the leaf length (*L*) and width (*W*) were measured using a tapeline of 0.1 cm and the maximum leaf area index (*LAImax*) was calculated using Equation (1):

$$LAImax = \sum_1^n L \times W \times 0.83 / A \tag{1}$$

where *n* is the number of all leaves, 0.83 is the leaf area computation coefficient for wheat and *A* is the land area of two 30 cm inner rows, which is always 1200 cm$^2$.

Similarly, at the summer maize silking stage, 5 representative maize plants with stable growth were randomly sampled in each plot. Then, the leaf length (*L*) and width (*W*) were measured for all leaves using a tapeline of 0.1 cm, and the maximum leaf area index (LAImax) was calculated using Equation (2):

$$LAImax = \sum_1^n L \times W \times 0.75 / A \tag{2}$$

where *n* is the number of all leaves, 0.75 is the leaf area computation coefficient for maize and *A* is the land area of 5 plants, which is always 5555 cm$^2$.

Additionally, at the spring wheat heading stage and summer maize silking stage, 50 flag (ear) leaf samples were randomly selected in each plot, and then, the maximum leaf chlorophyll content (SPADmax) was measured using a SPAD–502 Plus Minolta chlorophyll meter (Konica Minolta Inc., Tokyo, Japan).

### 2.3.4. Nitrogen Efficiency Index Calculation

The total plant N concentration (NC, %) was determined using $H_2SO_4$–$H_2O_2$ digestion and the micro–Kjeldahl procedure [46]. Additionally, 8 NUE measurements were calculated using Equations (3)–(10) [47–49]:

$$\text{Total N accumulation (TN, kg ha}^{-1}\text{)} = DM \times NC \tag{3}$$

$$\text{N use efficiency for biomass production (NUEbms, kg}^{-1}\text{)} = DM/TN \tag{4}$$

$$\text{N use efficiency for grain production (NUEg, kg}^{-1}\text{)} = GY/TN \tag{5}$$

$$\text{N harvest index (NHI)} = GY/DM \tag{6}$$

$$\text{N partial factor productivity (PFPN, kg}^{-1}) = GY_N/N_f \tag{7}$$

$$\text{N agronomy use efficiency (ANUE, kg}^{-1}) = (GY_N - GY_0)/N_f \tag{8}$$

$$\text{N recovery efficiency (NRE, \%)} = (TN_N - TN_0)/N_f \times 100 \tag{9}$$

$$\text{N uptake efficiency (NE}_{upk}, \text{kg}^{-1}) = TN_N/N_f \tag{10}$$

where DM and GY are the dry matter accumulation and grain yield for each plot, respectively; $GY_N$ and $GY_0$ are the grain yields in the N application plots and $N_0$ plots, respectively; $TN_N$ and $TN_0$ are TN in the N application plots and N0 plots, respectively; and $N_f$ is the applied amount of nitrogen fertilizer.

*2.4. Statistical Analysis*

Data were compiled using Microsoft Excel 2016. The treatment effects were analyzed using SPSS 22.0 (SPSS Inc., Chicago, USA) at the *p* = 0.05 level based on the least significant difference (LSD) method, and Pearson correlation analysis was used to analyze the correlation between GY and its components: DM, LAImax, SPADmax, and the nitrogen application rate, accumulation and utilization. A linear + platform model function was used to determine the relationship between the nitrogen application rate and spring wheat, summer maize, and annual grain yield using Simplot 10.0 (Systat, CA, USA) [33,34]. The functional formula was Equation (11):

$$GY = \begin{cases} a \times N + b & 0 < N < N_{opt} \\ c & N \geq N_{opt} \end{cases} \tag{11}$$

where *GY* is the spring wheat, summer maize and annual grain yield; *N* is the applied amount of nitrogen fertilizer; *a* and *b* are the equation coefficients; *c* is the platform yield, which always indicates the highest yield for the experiment; and $N_{opt}$ is the N application rate when *c* is reached, which always indicates the optimized N application rate.

**3. Results**

*3.1. Effect of Nitrogen Application Rate on Cereal Grain Yield and Yield Composition*

Compared with the $N_0$ treatment, the spring wheat yield increased significantly by 32.5–58.0% and 62.1–143.9% when the N application rate increased in 2018 and 2019, respectively (Figure 2). Meanwhile, the yields of the $N_{180}$ and $N_{360}$ treatments were significantly reduced compared to $N_{240}$ in 2018, but in 2019, the $N_{300}$ treatment obtained the highest yield, which was significantly higher than that of $N_{180}$ and $N_{240}$. Therefore, when the N application rate increased by 1 kg ha$^{-1}$, the spring wheat yield rapidly increased by 7.09 kg ha$^{-1}$ and 8.15 kg ha$^{-1}$ at first and then tended to remain flat with a linear–plateau tendency. The optimized N application rates were 231 kg ha$^{-1}$ and 307 kg ha$^{-1}$ in 2018 and 2019, respectively, and the corresponding platform yields were 5317 kg ha$^{-1}$ and 4654 kg ha$^{-1}$.

The summer maize yield under different N application rates was increased by 13.5–38.1% and 96.9–115.7% compared to the $N_0$ treatments in 2018 and 2019, respectively (Figure 2). The summer maize yield was the highest under the $N_{360}$ treatment, which increased by 15.1–21.6% compared to the $N_{180}$–$N_{300}$ treatments in 2018; however, there were no significant differences among treatments in 2019. The linear + platform function showed that summer maize yield increased linearly by 8.59 kg ha$^{-1}$ and 28.1 kg ha$^{-1}$ at first, with the N application rate increased by 1 kg ha$^{-1}$, and then obtained platform yields were 11,727 kg ha$^{-1}$ and 12,003 kg ha$^{-1}$ in 2018 and 2019, respectively, and the optimized N application rates were 337 kg ha$^{-1}$ and 222 kg ha$^{-1}$.

Due to the effects of the N application rate on the spring wheat and summer maize yields, the annual cereal grain yield was increased by 19.7–36.4% and 88.1–116.5% compared to the $N_0$ treatments in 2018 and 2019, respectively (Figure 2). Additionally, the annual cereal grain yield under the $N_{360}$ treatment was always lower than that of the $N_{480}$–$N_{720}$ treatments. With the increase in the N application rate by 1 kg ha$^{-1}$, the annual cereal grain yield linearly increased by 7.0 kg ha$^{-1}$ and 18.7 kg ha$^{-1}$ at first and then tended to remain

flat, and the highest yields were 16,658 kg ha$^{-1}$ and 16,349 kg ha$^{-1}$ at the annual optimized N application rates of 571 kg ha$^{-1}$ and 463 kg ha$^{-1}$ in 2018 and 2019, respectively.

Because the tilling time was short, the spring wheat tillers rarely developed into effective spikes, so there was no significant difference in the spike number in response to the different N application rates (Table 1). The spike grain number was significantly increased with the increase in the N application rate in 2019, but the 100–grain weight was always the highest under the $N_{240}$ treatment over the two years. There were no significant differences in the summer maize ears number among the different N application rates due to the use of the same plant density (Table 2). Additionally, the numbers of summer maize ear grains in response to the different N application rate treatments were significantly higher than those of the $N_0$ treatment, and the summer maize ear grain number of the $N_{180}$ treatment was lower than that of the $N_{360}$ treatment in 2018. In 2018, the summer maize 1000–grain weight under different N application rates was not significantly different, but with the increase in the N application rate, the summer maize 1000–grain weight significantly increased in 2019.

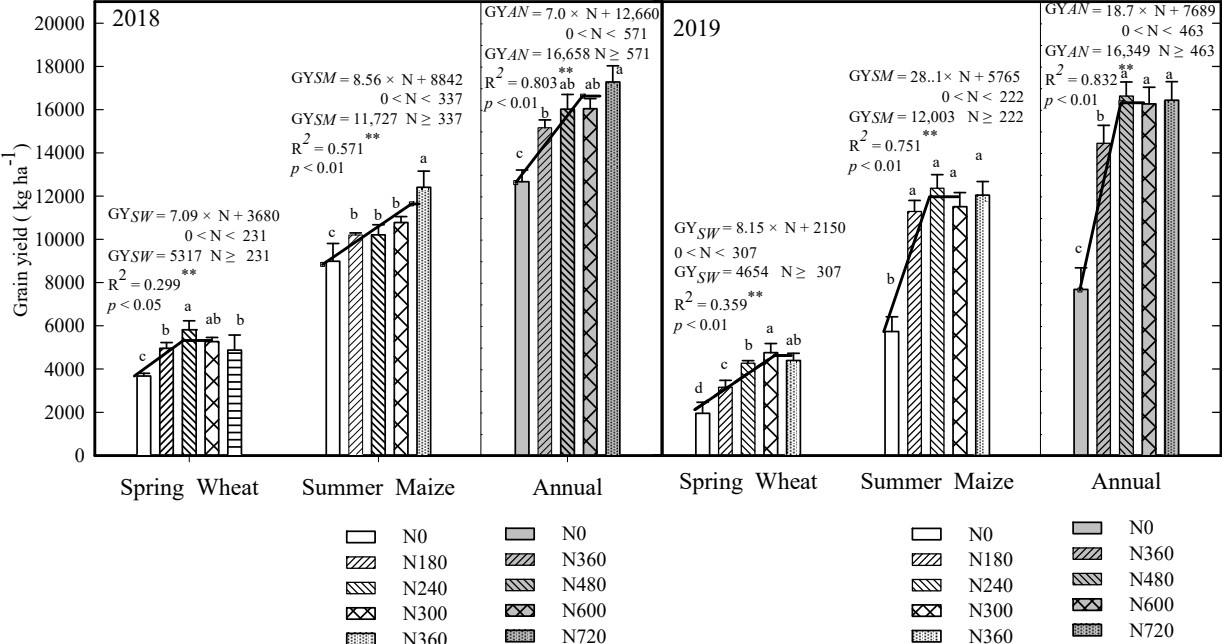

**Figure 2.** Spring wheat, summer maize and annual cereal grain yields in response to the N application rate. Note: The lowercase letters above bar graph indicate the significant differences at $p < 0.05$ levels, and the regression equation correlation coefficients are significant at ** = $p < 0.01$.

**Table 1.** Effect of the nitrogen application rate on the spring wheat grain yield composition.

| Years | Treatments | Spike Number (SN, $10^4$ ha$^{-1}$) | Spike Grain Number (SGN) | 100–Grain Weight (100–GW, g) |
|---|---|---|---|---|
| | N0 | 465 ± 30.3 [a] | 25.1 ± 3.1 [a] | 2.82 ± 0.13 [b] |
| | N180 | 440 ± 11.5 [a] | 25.5 ± 1.8 [a] | 2.94 ± 0.15 [ab] |
| 2018 | N240 | 476 ± 50.3 [a] | 25.4 ± 0.1 [a] | 3.13 ± 0.07 [a] |
| | N300 | 447 ± 18.8 [a] | 26.8 ± 0 [a] | 2.70 ± 0.11 [b] |
| | N360 | 448 ± 35.3 [a] | 27.9 ± 0.2 [a] | 3.36 ± 0.11 [a] |
| | N0 | 398 ± 79.4 [a] | 20.0 ± 1.6 [d] | 3.33 ± 0.15 [b] |
| | N180 | 403 ± 11.3 [a] | 24.7 ± 0.6 [c] | 3.81 ± 0.19 [a] |
| 2019 | N240 | 417 ± 38.6 [a] | 29.5 ± 0.7 [b] | 3.80 ± 0.09 [a] |
| | N300 | 443 ± 29.8 [a] | 31.1 ± 0.3 [ab] | 3.47 ± 0.14 [ab] |
| | N360 | 418 ± 57.2 [a] | 31.7 ± 0.8 [a] | 3.56 ± 0.11 [ab] |

Note: The lowercase letters after values indicate the significant differences at $p < 0.05$ levels.

### 3.2. Effect of Nitrogen Application Rate on Dry Matter Accumulation

Compared to the $N_0$ treatment, the spring wheat dry matter accumulation under different N application rates was significantly increased by 16.9–46.5% and 73.0–173.5% in 2018 and 2019, respectively (Figure 3). The dry matter accumulation of the $N_{360}$ treatment was significantly lower than that of the other N application rate treatments in 2018; however, in 2019, dry matter accumulation was significantly increased with the increase in the N application rate, and the dry matter accumulation of the $N_{360}$ treatment was significantly higher than that of the other treatments, while that of the $N_{180}$ treatment was the lowest ($p < 0.05$).

**Table 2.** Effect of the nitrogen application rate on the summer maize grain yield composition.

| Years | Treatments | Ear Number (EN, $10^4$ ha$^{-1}$) | Ear Grain Number (EGN) | 1000–Grain Weight (1000–GW, g) |
|---|---|---|---|---|
| 2018 | N0 | 9.0 [a] | 296.1 ± 17.2 [c] | 357.6 ± 12.2 [a] |
| | N180 | 9.0 [a] | 326.7 ± 16.4 [b] | 359.6 ± 21.6 [a] |
| | N240 | 9.0 [a] | 350.2 ± 12.1 [ab] | 353.8 ± 7.8 [a] |
| | N300 | 9.0 [a] | 334.8 ± 4.8 [ab] | 372.1 ± 25.5 [a] |
| | N360 | 9.0 [a] | 355.1 ± 8.1 [a] | 376.3 ± 23.4 [a] |
| 2019 | N0 | 9.0 [a] | 263.7 ± 46.4 [b] | 297.5 ± 5.7 [c] |
| | N180 | 9.0 [a] | 378.8 ± 27.3 [a] | 332.1 ± 17.1 [b] |
| | N240 | 9.0 [a] | 398.4 ± 41.9 [a] | 350.7 ± 6.8 [ab] |
| | N300 | 9.0 [a] | 389.1 ± 34.0 [a] | 358.3 ± 7.1 [a] |
| | N360 | 9.0 [a] | 415.0 ± 13.6 [a] | 352 ± 7.5 [ab] |

Note: The lowercase letters after values indicate the significant differences at $p < 0.05$ levels.

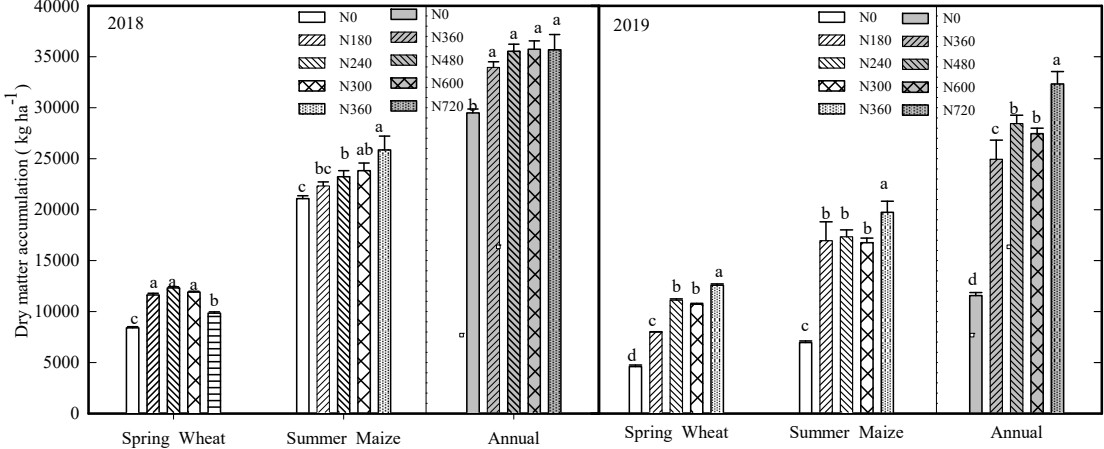

**Figure 3.** Spring wheat, summer maize and annual cereal dry matter accumulation responses to different nitrogen application rates. Note: The lowercase letters above bar graph indicate the significant differences at $p < 0.05$ levels.

The summer maize dry matter accumulation under different N application rates was significantly increased by 5.9–22.6% and 140.9–183.7% in 2018 and 2019, respectively, compared to the $N_0$ treatment. Over the two years, the dry matter accumulation significantly increased with the increase in the N application rate, and the dry matter accumulation of the $N_{360}$ treatment was significantly higher than that of the other treatments ($p < 0.05$).

Compared to the $N_0$ treatment, the annual cereal dry matter accumulation under different N application rates significantly increased by 15.1–21.2% and 115.6–179.7% in 2018 and 2019, respectively. There were no significant differences in the cereal dry matter accumulation among the N application rate treatments in 2018. In 2019, however, dry matter accumulation significantly increased with the increase in the N application rate, the dry mater accumulation of the $N_{360}$ treatment was significantly higher than that of the other treatments, while that of the $N_{180}$ treatment was the lowest ($p < 0.05$).

### 3.3. Effect of Nitrogen Application Rate on Maximum LAI and SPAD

The N application rate played significantly positive roles in the maximum leaf area index at the spring wheat heading stage and summer maize silking stage (Figure 4). Compared to the $N_0$ treatment, the spring wheat LAImax increased significantly by 16.4–41.2% and 23.9–33.6% with the increase in the N application rate in 2018 and 2019, respectively (Figure 4A). Meanwhile, the LAImax of the $N_{180}$ treatment was lower than that of the other treatments over the two years, and there were no significant differences between the $N_{300}$ and $N_{360}$ treatments.

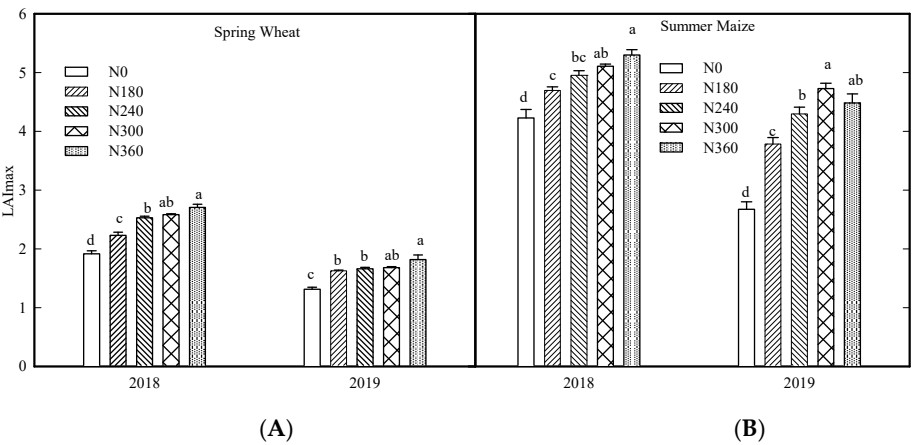

**Figure 4.** Effects of the nitrogen application rate on the spring wheat (**A**) and summer maize (**B**) maximum leaf area index at the flowering or silking stage. Note: The lowercase letters above bar graph indicate the significant differences at $p < 0.05$ levels.

Similar to spring wheat, the summer maize LAImax increased significantly by 11.1–25.4% and 41.5–76.8% with the increase in the N application rate compared to the $N_0$ treatment in 2018 and 2019, respectively (Figure 4B). Additionally, the LAImax under the $N_{180}$ treatment was significantly decreased compared to that of the $N_{300}$ and $N_{360}$ treatments over the two years, but the $N_{240}$ treatment's $LAI_{max}$ was significantly lower than that of the $N_{360}$ treatment in 2018 and the $N_{300}$ treatment in 2019; meanwhile, there were no significant differences between the $N_{300}$ and $N_{360}$ treatments.

There were no significant effects of the N application rate on the spring wheat flag leaf SPADmax in 2018 and 2019 (Figure 5A). With the increase in the N application rate, the summer maize ear leaf SPADmax was increased by 4.9–10.5% and 30.6–39.1% ($p < 0.05$) compared to that of the $N_0$ treatment in 2018 and 2019, respectively (Figure 5B). However, there were no significant differences among the different N application rate treatments.

Compared to the $N_0$ treatment, the spring wheat total N accumulation (TN) under the different N application rates was increased significantly by 30.7–50.1% and 96.0–270.1% in 2018 and 2019, respectively (Table 3). The TN under the $N_{180}$ and $N_{240}$ treatments was significantly higher than of the TN in the $N_{300}$ and $N_{360}$ treatments in 2018, while the TN of the $N_{360}$ treatment was significantly higher than that of the other treatments and that of the $N_{180}$ treatment was the lowest ($p < 0.05$) in 2019. There was a contrary tendency in the N use efficiency for biomass and grain production (NUEbms and NUEg), where the highest NUEbms and NUEg values were found under $N_{300}$ in 2018 and under $N_0$ in 2019; meanwhile, the lowest values were found under $N_{360}$ over the two years. In 2019, NUEbms and NUEg were decreased significantly by 11.8–26.1% and 14.6–41.9%, respectively, under the different N application rates compared to $N_0$. The N harvest index (NHI) did not show a significant difference among the different N application rates over the two years. In the spring wheat season, all treatments showed a decreasing tendency in N partial factor productivity (PFPN), N agronomy use efficiency (ANUE), N recovery efficiency (NRE) and N uptake efficiency (NEupk) with the increase in the N application rate in 2018, which were significantly increased by 29.5–103.3%, 58.8–167.8%, 36.0–226.9% and 23.8–129.8%,

respectively, compared to $N_{360}$ in 2018. In 2019, the highest PFPN, ANUE, NRE and NEupk values were found under $N_{240}$, which were 45.7%, 43.3%, 46.7% and 30.8% higher than the lowest values under $N_{360}$, $N_{180}$, $N_{180}$ and $N_{300}$, respectively.

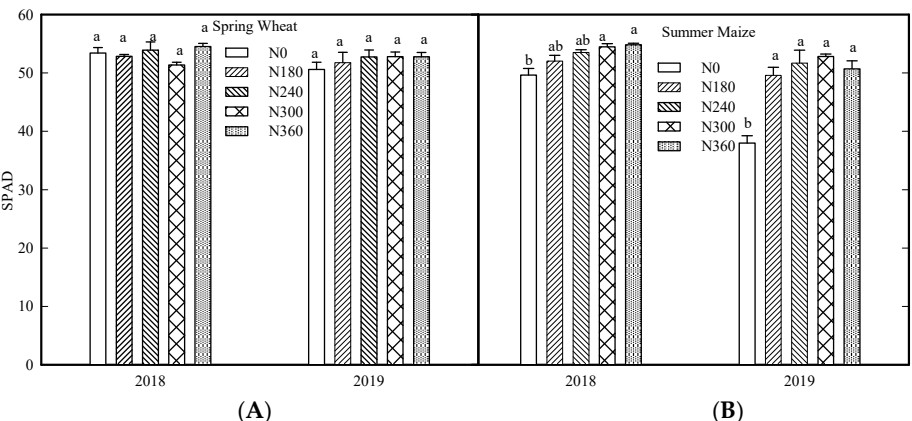

**Figure 5.** Effects of the nitrogen application rate on spring wheat (**A**) and summer maize (**B**) maximum leaf SPAD values at the flowering or silking stage. Note: The lowercase letters above bar graph indicate the significant differences at $p < 0.05$ levels.

*3.4. Effect of Nitrogen Application Rate on Spring Wheat, Summer Maize and Annual Cereal Yield Nitrogen Accumulation and Utilization*

In the summer maize season, TN was increased significantly with the increase in the N application rate (Table 3), increasing by 11.6–36.8% and 147.2–262.8% compared to the $N_0$ treatment in 2018 and 2019 ($p < 0.05$), respectively. The highest TN was found under $N_{360}$, which was 22.5% and 46.8% higher than the lowest values under $N_{180}$; meanwhile, there was a significant difference among the $N_{240}$, $N_{360}$ and $N_{180}$ treatments. Contrary to TN, the NUEbms and NUEg of summer maize under the different N application rates were decreased by 5.1–10.3% and 0.6–21.4% and 4.7–15.6% and 4.2–35.2% in 2018 and 2019, respectively, compared to the $N_0$ treatment. The highest NUEbms and NUEg were found under $N_0$, which were 10.3% and 21.4% and 15.6% and 35.2% higher than the lowest values under $N_{360}$. The NHI did not show a significant difference among the different N application rates over the two years for summer maize. The PFPN and NEupk of summer maize always showed a reducing tendency with the increase in the N application rate over the two years. $N_{180}$ had the highest PFPN and NEupk of 56.7 kg$^{-1}$–62.7 kg$^{-1}$ and 0.80 kg$^{-1}$–1.19 kg$^{-1}$ in 2018 and 2019, respectively, which were 64.3–87.4% and 36.3–63.3% higher than those of $N_{360}$, while $N_{240}$ had the second highest PFPN (42.5 kg$^{-1}$–51.5 kg$^{-1}$) and NEupk (0.71 kg kg$^{-1}$–0.98 kg$^{-1}$), with the values being 23.2–54.0% and 22.3–33.6% higher, respectively, than those of $N_{360}$. For the ANUE and NRE of summer maize, there was a contrary tendency over two years. The highest ANUE and NRE were found under $N_{360}$ in 2018, which were 87.3% and 58.2% higher than those of $N_{240}$ and $N_{180}$; however, in 2019, $N_{180}$ had the highest ANUE (30.9 kg$^{-1}$) and NRE (47.4%), with the values being 76.0% and 44,7% higher than those of $N_{360}$ and $N_{300}$, respectively.

For annual cereal production, when compared to $N_0$, all the treatments increased TN by 27.7–34.2% and 123.8–266.1% over the two years, and the TN under $N_{720}$ had the highest values, being 34.6% and 63.6% higher, respectively, than those of $N_{600}$ and $N_{360}$ in 2019. There was a significantly reduced tendency among all treatments for NUEbms and NUEg with the increase in the N application rate, where the highest NUEbms and NUEg of 108.2 kg kg$^{-1}$ and 71.9 kg kg$^{-1}$ were found under $N_0$ in 2019, with the values being 30.9% and 71.2% higher than those of $N_{360}$. There were also no significant differences among all treatments for the NHI. $N_{180}$ had the highest PFPN and NEupk of 40.1 kg$^{-1}$–42.1 kg$^{-1}$ and 0.66 kg$^{-1}$–1.18 kg kg$^{-1}$, which were 75.4–75.8% and 37.1–64.6% higher than those of $N_{360}$ and $N_{300}$, while $N_{240}$ had the second highest PFPN (33.4 kg$^{-1}$–34.6 kg$^{-1}$) and NEupk (0.65 kg$^{-1}$–0.91 kg$^{-1}$). The highest ANUE of 7.0 kg$^{-1}$– 18.8 kg$^{-1}$ was found under $N_{240}$, with the value being 24.2% and 53.3% higher

than that of $N_{300}$ and $N_{360}$ in 2018 and 2019, respectively. $N_{180}$ had the highest NRE (25.5%) in 2018, which was 15.2–61.9% ($p < 0.05$) higher than the NRE of the other treatments; however, the highest NRE (42.8%) was found under $N_{240}$ in 2019, with the value being 16.4% and 39.6% higher than that of $N_{180}$ and $N_{300}$, respectively.

*3.5. Relationships among Spring Wheat, Summer Maize and Annual Cereal GY and Its Components, DM, LAImax, SPADmax and Nitrogen Application Rate, Accumulation and Utilization*

Correlation analysis was performed to explore how the N application rate, accumulation and utilization affected the spring wheat GY and its components, DM, LAImax and SPADmax (Table 4). It was found that the N application rate played significantly positive roles in the spring wheat GY, SGN and DM, but there were no significant effects on SN and 100–GW, which indicates that N increased the spring wheat GY mainly by increasing SGN, since the increased N application rate always increased DM, which played significantly positive roles in GY and SGN, as well. The N application rate had positive effects on the spring wheat LAImax and SPADmax ($p > 0.05$), and the increased LAImax always increased GY through the positive effects on SN ($p < 0.05$) and DM ($p > 0.05$); however, LAImax played a negative role in 100–GW. Additionally, SPADmax always had a positive effect on GY ($p > 0.05$) due to the positive correlations between SPADmax, SN, SGN and DM, although the relationships were not significant. There was a positive relationship between the N application rate and TN ($p > 0.05$), but the other seven NUE measures, namely, NUEbm, NUEg, NHI, PFPN, ANUE, NRE and NEupk, always had a negative influence with the increase in the N application rate, and the correlation with PFPN was significant. There were significantly positive relationships between TN, GY, DM and LAImax, and the effects on SN, SGN and SPAD were positive but not significant, while the effects on 100–GW were negative and not significant. This indicates that, with the increase in the N application rate, TN increased, resulting in LAImax, SPADmax and DM increasing, allowing for SN and SGN formation and thus obtaining a higher GY. The effects of NUEbms, NUEg and NHI on SN, SGN, DM, LAImax and SPADmax were always negative, but on 100–GW, they were positive; therefore, NUEbms and NUEg had significantly negative effects on GY. However, the relationships between PFPN, ANUE, NRE and NEupk and GY, SN, SGN, 100–GW, DM, LAImax and SPADmax were not significant.

In the summer maize correlation analysis (Table 5), it was found that the N application rate played significantly positive roles in GY, EGN and 1000–GW, but there were no significant effects, which indicates that the summer maize GY was increased with the increase in the N application rate, mainly due to the increase in EGN and 1000–GW since the increased N application rate always increased DM, which played significantly positive roles in 1000–GW as well. The N application rate had significantly positive effects on the LAImax and SPADmax of summer maize, which always played significantly positive roles in 1000–GW, DM and EGN ($p > 0.05$); therefore, there were significant and positive relationships between GY, LAImax and SPADmax. There was a positive relationship between the N application rate and TN ($p > 0.05$), but the other seven NUE measures, namely, NUEbms, NUEg, NHI, PFPN, ANUE, NRE and NEupk, always showed a negative response to the N application rate, and the correlations for NUEbms and PFPN were significant. TN played significantly positive roles in the 1000–GW, DM, LAImax and SPADmax of summer maize, but the positive effects on GY were not significant. In contrast, NUEbms always had significant and negative effects on 1000–GW, DM, LAImax and SPADmax, and the negative roles in GY and EGN were not significant. The correlations among NUEg and NHI, and 1000–GW, DM, LAImax and SPADmax were positive, but NUEg and NHI had negative effects on EGN, which led to NUEg and NHI playing negative roles in GY. Similar to NUEbms, PFPN, ANUE and NRE always played significantly negative roles in 1000–GW, DM, LAImax and SPADmax; however, ANUE and NRE had significant and positive effects on EGN, which resulted in the positive relationships between GY and ANUE and NRE ($p > 0.05$). The effects of NEupk on GY were significantly negative due to the significant and negative relationships between NEupk and EGN.

**Table 3.** Spring wheat, summer maize and annual cereal nitrogen accumulation and utilization response to different nitrogen application rates.

| Crop | Year | N Application Rate | Total N Accumulation (TN, kg ha$^{-2}$) | N Use Efficiency for Biomass Production (NUEbms, kg$^{-1}$) | N Use Efficiency for Grain Production (NUEg, kg$^{-1}$) | N Harvest Index (NHI) | N Partial Factor Productivity (PFPN, kg$^{-1}$) | N Agronomy Use Efficiency (ANUE, kg$^{-1}$) | N Recovery Efficiency (NRE, %) | N Uptake Efficiency (NEupk, kg$^{-1}$) |
|---|---|---|---|---|---|---|---|---|---|---|
| Spring wheat | 2018 | N0 | 138.6 ± 7.3 [c] | 60.7 ± 1.6 [ab] | 25.3 ± 0.8 [ab] | 0.74 ± 0.012 [b] | | | | |
| | | N180 | 208.1 ± 10.1 [a] | 55.9 ± 1.6 [bc] | 25.4 ± 0.8 [ab] | 0.80 ± 0.013 [a] | 27.5 ± 1.1 [a] | 7.1 ± 0.5 [b] | 38.6 ± 2.7 [a] | 1.16 ± 0.08 [a] |
| | | N240 | 202.8 ± 4.0 [a] | 60.7 ± 2.7 [ab] | 27.6 ± 0.5 [a] | 0.76 ± 0.006 [ab] | 24.2 ± 2.1 [b] | 8.9 ± 0.5 [a] | 26.8 ± 1.6 [b] | 0.85 ± 0.05 [b] |
| | | N300 | 186.8 ± 4.3 [b] | 63.8 ± 1.9 [a] | 27.7 ± 0.9 [a] | 0.76 ± 0.012 [ab] | 17.5 ± 2.3 [c] | 5.3 ± 0.3 [c] | 16.1 ± 0.8 [c] | 0.62 ± 0.03 [c] |
| | | N360 | 181.1 ± 7.1 [b] | 54.3 ± 1.8 [c] | 24.7 ± 0.6 [b] | 0.81 ± 0.01 [a] | 13.5 ± 1.3 [d] | 3.3 ± 0.1 [d] | 11.8 ± 0.5 [d] | 0.5 ± 0.02 [d] |
| | 2019 | N0 | 48.9 ± 17.9 [d] | 94.1 ± 0.8 [a] | 46.9 ± 0.5 [a] | 0.81 ± 0.004 [a] | | | | |
| | | N180 | 95.9 ± 6.4 [c] | 83.1 ± 0.9 [b] | 40.1 ± 0.3 [b] | 0.78 ± 0.003 [a] | 17.6 ± 1.1 [a] | 6.7 ± 0.5 [b] | 26.1 ± 1.8 [b] | 0.53 ± 0.04 [ab] |
| | | N240 | 140.8 ± 15.6 [b] | 79.2 ± 0.8 [b] | 33.1 ± 0.3 [c] | 0.79 ± 0.004 [a] | 17.8 ± 2.0 [a] | 9.6 ± 0.6 [a] | 38.3 ± 2.3 [a] | 0.59 ± 0.04 [a] |
| | | N300 | 134.6 ± 12.3 [b] | 79.5 ± 0.7 [b] | 29.2 ± 0.3 [d] | 0.74 ± 0.004 [a] | 15.8 ± 2.3 [b] | 9.3 ± 0.5 [a] | 28.6 ± 1.5 [b] | 0.45 ± 0.02 [c] |
| | | N360 | 181.1 ± 16 [a] | 69.6 ± 0.7 [c] | 27.3 ± 0.3 [d] | 0.76 ± 0.004 [a] | 12.2 ± 1.3 [c] | 6.8 ± 0.3 [b] | 36.7 ± 1.4 [a] | 0.5 ± 0.02 [b] |
| Summer maize | 2018 | N0 | 193.3 ± 5.1 [d] | 109.1 ± 2.9 [a] | 49.9 ± 0.5 [a] | 0.70 ± 0.008 [ab] | | | | |
| | | N180 | 215.8 ± 9.6 [c] | 103.6 ± 3.2 [ab] | 46.3 ± 0.9 [ab] | 0.69 ± 0.013 [ab] | 56.7 ± 0.6 [a] | 6.8 ± 0.6 [b] | 12.5 ± 7.1 [c] | 1.19 ± 0.05 [a] |
| | | N240 | 235.5 ± 3.1 [b] | 98.6 ± 2.1 [ab] | 47.6 ± 1 [ab] | 0.71 ± 0.016 [a] | 42.5 ± 2.0 [b] | 5.1 ± 0.5 [c] | 17.6 ± 1.1 [b] | 0.98 ± 0.01 [b] |
| | | N300 | 242.5 ± 8.6 [ab] | 98.3 ± 3.5 [ab] | 44.1 ± 1.2 [bc] | 0.70 ± 0.019 [ab] | 35.9 ± 0.9 [c] | 6.0 ± 0.6 [bc] | 16.4 ± 1.4 [b] | 0.81 ± 0.03 [c] |
| | | N360 | 264.4 ± 15.1 [a] | 97.9 ± 3.1 [b] | 42.2 ± 1.9 [c] | 0.64 ± 0.029 [b] | 34.5 ± 2.1 [c] | 9.5 ± 1.0 [a] | 19.8 ± 1.9 [a] | 0.73 ± 0.04 [d] |
| | 2019 | N0 | 58.0 ± 3.3 [d] | 120.3 ± 6.7 [a] | 19.0 ± 0.3 [a] | 0.64 ± 0.018 [a] | | | | |
| | | N180 | 143.3 ± 18.2 [c] | 119.6 ± 14.6 [a] | 18.2 ± 0.6 [a] | 0.64 ± 0.021 [a] | 62.7 ± 6.2 [a] | 30.9 ± 0.6 [a] | 47.4 ± 4.8 [a] | 0.80 ± 0.10 [a] |
| | | N240 | 171.5 ± 13.3 [b] | 101.1 ± 1.9 [bc] | 16.2 ± 0.2 [b] | 0.67 ± 0.014 [a] | 51.5 ± 5.1 [b] | 27.6 ± 0.1 [b] | 47.3 ± 2.7 [a] | 0.71 ± 0.01 [b] |
| | | N300 | 156.3 ± 17.0 [bc] | 108.1 ± 11.8 [b] | 18.1 ± 0.3 [ab] | 0.62 ± 0.02 [a] | 38.4 ± 2.8 [c] | 19.3 ± 2.8 [c] | 32.8 ± 1.4 [c] | 0.52 ± 0.06 [d] |
| | | N360 | 210.3 ± 21.1 [a] | 94.5 ± 9.6 [c] | 12.3 ± 0.5 [c] | 0.63 ± 0.027 [a] | 33.5 ± 2.3 [c] | 17.5 ± 2.3 [d] | 42.3 ± 2.1 [b] | 0.58 ± 0.05 [c] |
| Annual | 2018 | N0 | 331.9 ± 22.6 [b] | 88.9 ± 6.1 [a] | 38.2 ± 2.6 [a] | 0.72 ± 0.05 [a] | | | | |
| | | N360 | 423.8 ± 17.8 [a] | 80.1 ± 3.4 [b] | 35.8 ± 1.5 [a] | 0.74 ± 0.03 [a] | 42.1 ± 1.8 [a] | 6.9 ± 0.3 [a] | 25.5 ± 1.07 [a] | 1.18 ± 0.05 [a] |
| | | N480 | 438.3 ± 24.5 [a] | 81.1 ± 4.5 [ab] | 36.5 ± 2.1 [a] | 0.73 ± 0.04 [a] | 33.4 ± 1.9 [b] | 7.0 ± 0.4 [a] | 22.2 ± 1.24 [b] | 0.91 ± 0.05 [a] |
| | | N600 | 429.2 ± 29.6 [a] | 83.3 ± 5.7 [ab] | 37.4 ± 2.6 [a] | 0.73 ± 0.05 [a] | 26.7 ± 1.4 [c] | 5.6 ± 0.4 [b] | 16.2 ± 1.12 [c] | 0.72 ± 0.05 [b] |
| | | N720 | 445.5 ± 13.8 [a] | 80.1 ± 2.5 [b] | 38.8 ± 1.2 [a] | 0.71 ± 0.02 [a] | 24.1 ± 0.7 [d] | 6.4 ± 0.2 [a] | 15.8 ± 0.49 [c] | 0.62 ± 0.02 [a] |
| | 2019 | N0 | 106.9 ± 7.3 [d] | 108.2 ± 7.4 [a] | 71.9 ± 4.9 [a] | 0.72 ± 0.05 [a] | | | | |
| | | N360 | 239.2 ± 10.0 [c] | 104.2 ± 4.4 [a] | 60.4 ± 2.5 [b] | 0.70 ± 0.03 [a] | 40.1 ± 1.6 [a] | 18.8 ± 0.8 [a] | 36.8 ± 1.5 [b] | 0.66 ± 0.03 [a] |
| | | N480 | 312.3 ± 17.5 [b] | 91.1 ± 5.1 [b] | 53.2 ± 3.0 [c] | 0.73 ± 0.04 [a] | 34.6 ± 1.9 [b] | 18.6 ± 1.0 [a] | 42.8 ± 2.4 [a] | 0.65 ± 0.04 [a] |
| | | N600 | 290.8 ± 20.1 [b] | 94.4 ± 6.5 [b] | 55.9 ± 3.9 [bc] | 0.68 ± 0.05 [a] | 27.1 ± 1.2 [c] | 14.1 ± 1.0 [b] | 30.7 ± 2.1 [c] | 0.48 ± 0.03 [c] |
| | | N720 | 391.3 ± 12.1 [a] | 82.6 ± 2.6 [c] | 42.0 ± 1.3 [d] | 0.69 ± 0.02 [a] | 22.8 ± 0.6 [d] | 12.25 ± 0.4 [c] | 39.5 ± 1.2 [ab] | 0.54 ± 0.02 [b] |

Note: The lowercase letters after values indicate the significant differences at $p < 0.05$ levels.

**Table 4.** The correlations between GY and its components, DM, LAImax and SPADmax and nitrogen accumulation and utilization of spring wheat.

| | N | GY | SN | SGN | 100–GW | DM | LAImax | SPADmax |
|---|---|---|---|---|---|---|---|---|
| N | 1 | | | | | | | |
| GY | 0.689 * | 1 | | | | | | |
| SN | −0.091 | 0.503 | 1 | | | | | |
| SGN | 0.791 ** | 0.552 | −0.284 | 1 | | | | |
| 100–GW | 0.241 | −0.332 | −0.773 ** | 0.302 | 1 | | | |
| DM | 0.745 * | 0.904 ** | 0.235 | 0.733 * | −0.138 | 1 | | |
| LAImax | 0.492 | 0.802 ** | 0.813 ** | 0.146 | −0.554 | 0.574 | 1 | |
| SPADmax | 0.388 | 0.616 | 0.473 | 0.445 | −0.042 | 0.483 | 0.569 | 1 |
| TN | 0.607 | 0.925 ** | 0.592 | 0.486 | −0.419 | 0.894 ** | 0.821 ** | 0.631 |
| NUEbms | −0.335 | −0.746 * | −0.820 ** | −0.238 | 0.589 | −0.605 | −0.874 ** | −0.728 * |
| NUEg | −0.495 | −0.823 ** | −0.575 | −0.581 | 0.462 | −0.766 ** | −0.728 * | −0.760 * |
| NHI | −0.076 | −0.312 | −0.045 | −0.414 | 0.242 | −0.364 | −0.007 | −0.135 |
| PFPN | −0.769 * | 0.382 | 0.531 | −0.673 | −0.471 | 0.211 | 0.233 | 0.017 |
| ANUE | −0.417 | −0.001 | −0.431 | 0.217 | 0.322 | 0.233 | −0.605 | −0.109 |
| NRE | −0.404 | −0.215 | −0.471 | 0.259 | 0.319 | 0.325 | −0.645 | −0.133 |
| NEupk | −0.607 | 0.422 | 0.584 | −0.578 | −0.556 | 0.355 | 0.334 | 0.092 |

Note: Correlation coefficients are significant at * = *p* < 0.05 and ** = *p* < 0.01. N, N application rate; GY, grain yield; SN, spike number; SGN, spike grain number; 100–GW, 100–grain weight; DM, dry matter accumulation; LAImax, maximum leaf area index; SPADmax, maximum SPAD value; TN, total N accumulation; NUEbms, N use efficiency for biomass production; NUEg, N use efficiency for grain production; NHI, N harvest index; PFPN, N partial factor productivity; ANUE, N agronomy use efficiency; NRE, N recovery efficiency; NEupk, N uptake efficiency.

**Table 5.** The correlations between GY and its components, DM, LAImax and SPADmax, and nitrogen accumulation and utilization of summer maize.

| | N | GY | EGN | 1000–GW | DM | LAImax | SPADmax |
|---|---|---|---|---|---|---|---|
| N | 1 | | | | | | |
| GY | 0.841 ** | 1 | | | | | |
| EGN | 0.783 ** | 0.887 ** | 1 | | | | |
| 1000–GW | 0.640 * | 0.716 * | 0.386 | 1 | | | |
| DM | 0.552 | 0.621 | 0.275 | 0.924 ** | 1 | | |
| LAImax | 0.740 * | 0.719 * | 0.433 | 0.963 ** | 0.924 ** | 1 | |
| SPADmax | 0.732 * | 0.834 ** | 0.577 | 0.939 ** | 0.896 ** | 0.958 ** | 1 |
| TN | 0.632 | 0.632 | 0.311 | 0.918 ** | 0.984 ** | 0.937 ** | 0.881 ** |
| NUEbm | −0.731 * | −0.618 | −0.461 | −0.772 ** | −0.746 * | −0.816 ** | −0.728 * |
| NUEg | −0.143 | −0.127 | −0.495 | 0.522 | 0.673 * | 0.504 | 0.391 |
| NHI | −0.264 | −0.167 | −0.382 | 0.313 | 0.473 | 0.299 | 0.278 |
| PFPN | −0.935 ** | −0.287 | −0.118 | −0.691 | −0.438 | −0.729 * | −0.627 |
| ANUE | −0.232 | 0.545 | 0.741 * | −0.740 * | −0.868 ** | −0.861 ** | −0.753 |
| NRE | −0.038 | 0.617 | 0.887 ** | −0.720 * | −0.830 * | −0.813 * | −0.759 * |
| NEupk | −0.667 | −0.759 * | −0.792 * | 0.027 | 0.446 | 0.105 | 0.086 |

Note: Correlation coefficients are significant at * = *p* < 0.05 and ** = *p* < 0.01. N, N application rate; GY, grain yield; EGN, ear grain number; 1000–GW, 1000–grain weight; DM, dry matter accumulation; LAImax, maximum leaf area index; SPADmax, maximum SPAD value; TN, total N accumulation; NUEbms, N use efficiency for biomass production; NUEg, N use efficiency for grain production; NHI, N harvest index; PFPN, N partial factor productivity; ANUE, N agronomy use efficiency; NRE, N recovery efficiency; NEupk, N uptake efficiency.

For annual cereal yield, the N application rate played significantly positive roles in GY and DM but had significant and negative effects on PFPN, and the positive relationship between the N application rate and TN was not significant (Table 6). Annual cereal DM and TN were always significantly positively related to GY, while GY was negatively affected by NUEbms and PFPN ($p > 0.05$). The relationships between DM and TN were significant and positive, and NUEbms, NUEg, ANUE and NRE played significantly negative roles in DM.

**Table 6.** The correlations between annual cereal GY, DM and nitrogen accumulation and utilization.

| | N | GY | DM | TN | NUEbms | NUEg | NHI | PFPN | ANUE | NRE | NEupk |
|---|---|---|---|---|---|---|---|---|---|---|---|
| N | 1 | 0.868 ** | 0.654 * | 0.631 | −0.564 | −0.381 | −0.342 | −0.985 ** | −0.294 | −0.224 | −0.636 |
| GY | | 1 | 0.853 ** | 0.789 ** | −0.703 * | −0.609 | −0.145 | −0.779 * | −0.255 | −0.201 | −0.437 |
| DM | | | 1 | 0.984 ** | −0.926 ** | −0.925 ** | 0.175 | −0.338 | −0.946 ** | −0.760 * | 0.432 |

Note: Correlation coefficients are significant at * = $p < 0.05$ and ** = $p < 0.01$. N, N application rate; GY, grain yield; DM, dry matter accumulation; TN, total N accumulation; NUEbms, N use efficiency for biomass production; NUEg, N use efficiency for grain production; NHI, N harvest index; PFPN, N partial factor productivity; ANUE, N agronomy use efficiency; NRE, N recovery efficiency; NEupk, N uptake efficiency.

## 4. Discussion

Numerous studies have shown that the grain yield of crops can be improved by N application [9], and N application contributes to about 30–50% of increases in crop grain yield [2], especially in the higher–annual–yielding region of the North China Plain [19,50–52]. The results indicate that the N treatments always increased spring wheat, summer maize and annual grain yield by 32.5–143.9%, 13.5–115.7% and 19.7–116.5%, respectively (Figure 2). Meanwhile, the grain yield in the $N_{240}$ treatment (5038 kg ha$^{-1}$ and 11,282 kg ha$^{-1}$) and $N_{480}$ treatment (16320 kg ha$^{-1}$) averaged over two years were the highest for spring wheat, summer maize and annual grain, respectively. Generally, a linear–plateau model identified the optimal N application for maximum crop yields [53]. In this study, we estimated an optimal N application rate of 231–307 kg ha$^{-1}$, 222–337 kg ha$^{-1}$ and 463–571 kg ha$^{-1}$ for spring wheat, summer maize and annual cereal yield, respectively, above which grain yields did not appear to increase with N application rates; the maximum grain yields were 4654–5317 kg ha$^{-1}$, 11,727–12,003 kg ha$^{-1}$ and 16,349–16,658 kg ha$^{-1}$ for spring wheat, summer maize and annual cereal yield, respectively (Figure 1). Approximately 65–80% of the observations were based on crop fields receiving N fertilizer above the optimal N application rate, which were higher than the rate used in winter wheat–summer maize annual systems in the North China Plain [53–55] and were lower than others reported for wheat and maize production [56,57]. The differences are probably a result of the different N requirements between spring wheat and winter wheat because spring wheat has a shorter growth duration without overwintering, and the summer maize growth duration in this study was 30 days longer than the traditional harvest date in order to obtain reduced grain moisture contents.

Cereal grain yield is mainly determined by spike (ear) number, grains per m$^2$ or grain weight, while changes in grains m$^{-2}$ are primarily associated with spike number m$^{-2}$ [58,59]. However, in this study, the N application rate played no significant effects on spring wheat spike number (Table 1), which were different than winter wheat mainly due to the tilling time of spring wheat being shorter and the tillers rarely developing into effective spikes [60]. Optimized N application rate and the ratio of different stage could significantly increase the winter wheat spike number and spike grains number. The 1000–grain weight showed no difference [61]. This study confirmed that the spring wheat spike grain number was significantly increased with the increase in the N application rate, but the 100–grain weight for $N_{240}$ treatment was the highest. A recent study showed that maize grain yield increased by 2.7–10.5% in China without extra nitrogen input due to increased ear number as a result of increased plant density [55]. Plant density thus not only affected the yield of ear number per area but also the number of ear kernels per kernel weight [62]. In this study, the number of ear grains and the 1000–grain weight of summer maize were increased significantly by an increased N application rate. Furthermore, correlation analysis showed that the N application rate played significantly positive roles in the GY of spring wheat and summer maize (Tables 4 and 5), increasing spring wheat SGN and DM and summer maize EGN and 1000–GW. These findings indicate that N increased DM, which played significantly positive roles on spring wheat SGN and summer maize 1000–GW, as well. The N application rate played significantly positive roles on annual cereal DM and TN, which were always significantly positively related to GY, but GY was negatively affected by NUEbms and PFPN ($p>0.05$) (Table 6).

To date, numerous studies have shown that the accumulated DM and its allocation into kernels is the material basis for yield formation, which contributes to 73.7% of higher grain

yields [63,64]. In this study, spring wheat, summer maize and annual cereal DM accumulation under different N application rates were significantly increased by 16.9–173.5%, 5.9–22.6%, and 140.9–183.7% and 15.1–179.7%, respectively, compared to the $N_0$ treatment (Figure 3). Meanwhile, the DM accumulation among different N treatments was also significantly increased with the increase in the N application rate over two years. The effects of the N application rate on the grain yield difference were more obvious than the effects of DM, especially when N application was higher than the $N_{240}$ treatment (Figures 2 and 3). These results indicate that the grain yield always improved mainly due to the greater DM and higher harvest index [62,65–68]. These results were confirmed by Meng et al. [69], who found that wheat and maize grain yield increase to 6.0–7.5 and 10.5–12.0 $10^3$ kg ha$^{-1}$, respectively, as a result of increasing DM and harvest index. Using correlation analysis, the authors showed that N application rate exerted significantl effects on spring wheat DM, and positively affected TN, LAImax and SPADmax ($p > 0.05$), but negatively affected NUEg (Table 4). Moreover, LAImax, SPADmax, TN and NUEg were significantly and positively affected by summer maize DM, while NUEbms, ANUE and NRE played significantly negative roles (Table 5). For annual cereal yield, the N application rate and TN played significantly positive roles in DM, which had significantly negative correlations with NUEbms, NUEg, ANUE and NRE (Table 6). These findings agree with previous studies reporting that increased DM, which is determined by LA and SPAD, affects crop growth and nutritional status and is thus a significant basis for GY formation [70–73].

A recent study found that N fertilization management affects LAI and SPAD, which can assess the N nutrition status of crops [71,74–77] and thus provide guidance for more accurate N management [78,79]. The present results indicate that the maximum LAIs (LAImax) of spring wheat and summer maize were significantly and positively affected by N application rate (Figure 4); these were increased by 16.4–41.2% and 11.1–76.8% over $N_0$ treatment, respectively, and the highest LAImax of 4.20–5.11was observed under the $N_{300}$ treatment. Therefore, N application rate and TN had significantly relationship with the LAImax of spring wheat and summer maize, which in turn had a negative relationship with NUEbms, NUEg, ANUE and NRE (Tables 4 and 5). These findings are supported by previous studies that an LAI exceeding 5.0 is not necessary for obtaining a high grain yield [65,80], and an LAImax of about 5.0–6.0 might be optimal for maize [81,82]: even an LAImax of 6.7 for maize grain yield to exceed 20 Mg ha$^{-1}$ [83], so plant N uptake is thus proportional to LAI [84]. However, in this study, it is interesting that spring wheat flag leaf and summer maize's maximum SPAD (SPADmax) did not show significant differences in response to different N application rates; these results differ from other reports that state SPAD was higher in optimized N treatments [61], a discrepancy which might be due to the higher soil foundation fertility. Our correlation analysis showed that N application rate and TN always had significantly positive effects on the SPADmax of spring wheat and summer maize, which had negative relationship with NUEbms, NUEg, PFPN, ANUE and NRE (Tables 4 and 5). In this study, increased N application increased TN and exerted positive effects on LAImax and SPADmax and increased the DM material base for SGN/EGN and GW, increasing the overall GY at end of the growing period. The results were supported by recent studies that crop management can coordinate with Nupk by improving the canopy's eco–physiological characteristics, resulting in higher yield and NUE [85,86].

It is believed that N application could increase wheat and summer maize grain yield, but we found that yield was not increased linearly [2,11]. Numerous studies observed that improving N uptake ability is the first step to increase N use efficiency, which refers to the N management level and the plants' nitrogen absorption converted to grain yield [87–89]. The present results indicated that the TN under $N_{360}$ treatments always significantly increased by 5.4–19.1% and 16.6–32.3% for spring wheat and summer maize, respectively, and by 11.5–26.2% under $N_{720}$ treatments for annual cereal yield compared to other treatments; moreover, the $N_{360}$ and $N_{720}$ significantly increased TN by 30.7–270.1%, 11.6–262.8% and 27.7–266.1% for spring wheat, summer maize and annual cereal yield compared to the $N_0$ treatment, respectively, over the two years (Table 3), which is in agreement with previous studies. However, increased TN under different N treatments lead to reduced N use

efficiency in this study. The NUEbms and NUEg for the $N_{360}$ treatment were always significantly decreased compared to $N_{180}$–$N_{300}$ by 10.9–13.6% and 8.9–20.7%, 6.8–13.8% and 12.2–15.6%, for spring wheat and summer maize, respectively, over two years. In addition, annual cereal yield under the $N_{720}$ treatment was reduced by 5.5–11.7% and 10.0–16.0% compared to the others. NHI did not show a significant difference among all treatments over the two years. Furthermore, it is believed that the N use efficiency always decreased with increasing N application rates [90], even leading to 26–31% lower N use efficiency in the wheat–maize crop rotation system in the North China Plain [22,25]. In this study, in the spring wheat and summer maize season, all treatments showed a decreasing tendency in PFPN, ANUE, NRE and NEupk with the increase in the N application rate. Under the $N_{240}$ treatment, the PFPN, ANUE, NRE and NEupk for spring wheat obtained high levels of 20.99 kg$^{-1}$, 9.27 kg$^{-1}$, 32.53% and 0.72 kg$^{-1}$ over two years; meanwhile those for summer maize were 47.01 kg$^{-1}$, 16.35 kg$^{-1}$, 32.44% and 0.85 kg$^{-1}$, respectively, which were not different from $N_{180}$ and which were significantly higher than the $N_{300}$ and $N_{360}$ treatments. However, in annual crops systems, the PFPN and NEupk for $N_{360}$ were significantly higher than others. In addition, the ANUE and NRE for $N_{360}$ and $N_{480}$ were significantly higher than $N_{600}$ and $N_{720}$. NEupk reflects the level of N management [63]. In this study, the NEupk values were 0.42–1.19, which was similar to the international recommended value of 0.5–0.9 [91]. These findings were supported by Han et al. [92], who highlighted that an NEupk < 1 indicates that the uptake of N is less than the input N. In this study, the N application rate of $N_{600}$ and $N_{720}$ were wasted under the annual spring wheat–summer maize cropping system.

## 5. Conclusions

Our study demonstrated that, with an increase in the N application rate, the LAImax of spring wheat and summer maize significantly increased by 16.4–41.2% and 5.9–183.7%, respectively, which resulted in dry matter accumulation increasing by 16.9–173.5% and 11.1–76.8%, respectively, and the annual cereal dry matter accumulation was 15.1–179.7%. Spring wheat, summer maize and annual cereal obtained higher grain yield of 5038, 1282 and 16320 kg ha$^{-1}$, respectively, under $N_{240}$ treatment. The total N accumulation and N uptake efficiency in the crops were increased with an increased N application rate, which a played positive role in grain yield formation; however, N use efficiency for biomass and grain production, N partial factor productivity, N agronomy use efficiency and N recovery efficiency were decreased significantly and negatively affected grain yield formation. Thus, due to the positive effects of the N application rate on the maximum LAI and SPAD, and dry matter accumulation, spring wheat spike grain number and summer maize ear grain number and grain weight increased, yielding higher grain yield and N accumulation and utilization. We obtained the optimal N application rates of 231–307, 222–337 and 463–571 kg ha$^{-1}$ for spring wheat, summer maize and annual cereal, respectively, using a linear–plateau model. The highest grain yields were 4654–5317, 11727–12,003 and 16,349–16,658 kg ha$^{-1}$, respectively. These results demonstrate the theoretical basis for transitioning from winter wheat to spring wheat in a wheat–summer maize crop system to improve annual mechanical harvesting and identified the rational and efficient N fertilizer application rate, and which provide N application guidance to farmer in the thermal–resource–limited region of North China Plain.

**Author Contributions:** Funding acquisition and conceiving and designing the experiment, M.L., Y.Y., H.H. and J.G.; data collection and writing—original draft, M.L., Z.M., Q.L., Y.Z. and Y.Y.; writing—review and editing, M.L, X.W. and J.G. All authors have read and agreed to the published version of the manuscript.

**Funding:** This work was funded by National Natural Science Foundation of China (31701378), and the Key National Research and Development Program of China (2017YFD0300305).

**Data Availability Statement:** Not applicable.

**Conflicts of Interest:** The authors declare no conflict of interest.

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
