# Peer review of "Spring Wheat–Summer Maize Annual Crop System Grain Yield and Nitrogen Utilization Response to Nitrogen Application Rate in the Thermal–Resource–Limited Region of the North China Plain"

_agronomy, doi:10.3390/agronomy13010155_

Round 1
Reviewer 1 Report
Comments to the Authors
The manuscript is well written, the present study compared the effects Spring wheat-summer maize annual crop system grain yield and nitrogen utylization respond to nitrogen application rate in the thermal-resource-limited Region of the North Plain.
Other minor comments are described below.
L146: N (300), N (360) kg ha-1 These are very high doses of N, unacceptable in Poland, because the accumulation of nitrates in plants. Dangerous for animals and humans. It would be good to check if N does not accumulate in the plant. There can also be losses of N in the soil and pollution of the environment.
Even more was used for annual crops, there is a lot in Europe.
Conclusion
line 3 11.1% - 76.8% removing the mark.
After minor corrections, the manuscript can be published.
Author Response
Reviewer #1: Comments to the Authors
The manuscript is well written, the present study compared the effects Spring wheat-summer maize annual crop system grain yield and nitrogen utylization respond to nitrogen application rate in the thermal-resource-limited Region of the North Plain.
Other minor comments are described below.
L146: N (300), N (360) kg ha-1 These are very high doses of N, unacceptable in Poland, because the accumulation of nitrates in plants. Dangerous for animals and humans. It would be good to check if N does not accumulate in the plant. There can also be losses of N in the soil and pollution of the environment.
Even more was used for annual crops, there is a lot in Europe.
----R: Thanks for your helpful suggestion. In this manuscript (MS) we set the higher N application rate as N (300), N (360) kg ha-1 to find the N application rate optimized point on spring wheat and summer maize, while farmers always applied higher N levels than N (300) kg ha-1 in production. And as the reviewer’s suggestion we studied the N cycle in 2020-2022, which will be written in another MS.
Conclusion
line 3 11.1% - 76.8% removing the mark.
----R: Thanks for your helpful suggestion. We remove the mark in the revised MS.
After minor corrections, the manuscript can be published.
----R: Thanks for your constructive comments for our MS.
Other changes: We do not list other changes here but have marked them in red in the revised manuscript.
In all, the comments of editor and reviewers are all helpful to us, and we have revised the paper point by point. Thanks for your insightful comments.
We look forward to hearing from your responses at your earliest convenience.
Best wishes,
Junzhu Ge
22nd Nov., 2022

Reviewer 2 Report
The subject matter is good, but this piece needs serious English editing to make it easier to read than it currently is. Some of the sentences are way too long and confusing.

Author Response
----R: Thanks for your helpful suggestion. We have edited the English carefully, and were modified by MDPI Billing Team (https://susy.mdpi.com/user/pre_english_article/status- MDPI English editing-54440)
Other changes: We do not list other changes here but have marked them in red in the revised manuscript.
In all, the comments of editor and reviewers are all helpful to us, and we have revised the paper point by point. Thanks for your insightful comments.
We look forward to hearing from your responses at your earliest convenience.
Best wishes,
Junzhu Ge
22nd NOV., 2022

Reviewer 3 Report
The manuscript, entitled " Spring Wheat–Summer Maize Annual Crop System Grain Yield and Nitrogen Utilization Respond to Nitrogen Application Rate in the Thermal-Resource-Limited Region of the North China Plain", Meng Liu, presents results related to the effect of 5 N fertilization levels on wheat, summer maize and annual crops yields.
The manuscript doesn’t respond to the main criteria of a research paper. The text is difficult to read due to a high degree of detail and a high usage of abbreviations. There are many imperfections mainly in the statistical analysis of the data and the writing needs a major revision. Generally, this study is only an experiment report without any new findings or contributions.
The abstract jumps in the middle of the methodology, but it should first have a brief statement of the context and the problem that the study addresses. Without that, readers are lost.
There are many details and many abbreviations that mean hard work for a reader who tries to make sense of the findings. Suggest considering omitting some detail, letting the reader look in the tables and figures for details, and just reporting the main trends in the data, with less numerical information, and more emphasis on results that are of broader relevance for the interpretation of the study results. Make sparing use of abbreviations.
Author Response
The manuscript doesn’t respond to the main criteria of a research paper. The text is difficult to read due to a high degree of detail and a high usage of abbreviations. There are many imperfections mainly in the statistical analysis of the data and the writing needs a major revision. Generally, this study is only an experiment report without any new findings or contributions.
----R: Thanks for your strictness review and helpful suggestion. (1) We have modified the manuscript (MS) into a good reading experience by deleted more detail, and changed abbreviations into full title. (2) We perfected the data statistical analysis in the revised MS, and we have edited the English carefully, and were modified by MDPI Billing Team. (https://susy.mdpi.com/user/pre_english_article/status- MDPI English editing-54440)). And (3) we enhanced the MS academic level and makes it more like a research paper, which can provide a theoretical basis of the N application rational and efficient, and guide farmers to apply N for spring wheat-summer maize annual crop system in the thermal-resource-limited region of North China Plain.
The abstract jumps in the middle of the methodology, but it should first have a brief statement of the context and the problem that the study addresses. Without that, readers are lost.
----R: Thanks for your helpful suggestion. We have modified the MS abstract, highlight the context and the problem for this study. Please see the revised manuscript with red changes marked.
There are many details and many abbreviations that mean hard work for a reader who tries to make sense of the findings. Suggest considering omitting some detail, letting the reader look in the tables and figures for details, and just reporting the main trends in the data, with less numerical information, and more emphasis on results that are of broader relevance for the interpretation of the study results. Make sparing use of abbreviations.
----R: Thanks for your strictness review and helpful suggestion. We have modified the MS into a good reading experience by deleted more detail, and changed abbreviations into full title.
Other changes: We do not list other changes here but have marked them in red in the revised manuscript.
In all, the comments of editor and reviewers are all helpful to us, and we have revised the paper point by point. Thanks for your insightful comments.
We look forward to hearing from your responses at your earliest convenience.
Best wishes,
Junzhu Ge
22nd NOV., 2022

Round 2
Reviewer 3 Report
The manuscript has been improved mainly in English and readability. I particularly appreciate the authors' feedback. However, the manuscript still needs extensive revision to improve its scientific soundness and context. My main concerns with this manuscript are the following.
1. Climatic conditions and principally rainfall are known to be major factors determining the yield productivity of these crops. In the same way, the management of nitrogen fertilization is highly dependent on climatic conditions. However, this aspect is not considered in the present manuscript at all. In addition, an application of irrigation was shown in the presented experiment. But I do not understand the adopted irrigation calendar. The statement "50 mm for wheat and 75 mm for maize each time" is not informative! !! Total rainfall and total water used for irrigation by crop season and crop are very important parameters. Unfortunately, they are not given and discussed in the manuscript.
2. Nitrogen is a key factor determining the yield of these crops. However, nitrogen fertilization is highly dependent on soil nitrogen content and organic matter. The determination of crop N fertilization requirements is related to soil organic matter. However, the authors do not consider this main factor in the manuscript. In such an experiment, soil organic matter (at least) should be analyzed before sowing and after harvest.
3. The statistical analysis of the data needs to be clarified and corrected. Here, the experimental design is an RCBD with nitrogen fertilization as the only experimental factor and the experiment was conducted for two seasons. I do not understand why each season was analyzed separately. Also, the choice of the post-hoc test must be justified. In such an experiment, Tukey's test is considered the appropriate post hoc test and not the LSD. The ANOVA assumption tests must be reported. The ANOVA output table should be added to the manuscript. There was a significant interaction between season and nitrogen fertilization factors?
For the above reasons, the presented manuscript does not satisfy the threshold of maturity to be published. I understand the effort made by the authors to improve their manuscript, but I insist that the present manuscript needs a thorough revision to be considered in the future. I also note that other reviewers give positive feedback on this study. I can understand other different evaluations and accept any final decision made by the editor.
Author Response
1. Climatic conditions and principally rainfall are known to be major factors determining the yield productivity of these crops. In the same way, the management of nitrogen fertilization is highly dependent on climatic conditions. However, this aspect is not considered in the present manuscript at all. In addition, an application of irrigation was shown in the presented experiment. But I do not understand the adopted irrigation calendar. The statement "50 mm for wheat and 75 mm for maize each time" is not informative!!! Total rainfall and total water used for irrigation by crop season and crop are very important parameters. Unfortunately, they are not given and discussed in the manuscript.
----R: Thanks for your strictness review. Firslty, in this MS, we mainly consider N application rate as the only artificial control factor to study the spring wheat and summer maize yield formation and N use efficiency. And so we did not consider the different effects of climatic conditions and rainfall, which just as the reviewer said be major factors determining the yield productivity, and that we will be fully considered in our future studies. Secondly, the reviewer’s attention of “ irrigation calendar”, the cropping management of “irrigated with 50 mm for wheat and 75 mm for maize in different stage” were mainly based on the farmers management measures, and we did not consider the water influence on the spring wheat and summer maize as the splits at the same levels.
And in order to put away review worries, we calculated the growth degree days and total rainfall during spring wheat and summer maize growth duration, and added in the 2.1 sections in the revised MS as belows:
The growth degree days for spring wheat were 1907.5 and 1923.6 ℃d in 2018 and 2019, with total rainfull be 67.0 and 89.4 mm, respectively. And during summer maize seasons the growth degree days and total rainfull were 2871.2 and 2880.7 ℃d, and 387.7 and 344.5 mm, respectively.
And in the revised MS, in the 2.2 section, were changes marked as described below.
…The field was irrigated with 50 mm each time in the elongation and heading stages based on the farmers management measures…. The field irrigation was 75 mm each time in the jointing and silking stages based on the farmers management measures.
2. Nitrogen is a key factor determining the yield of these crops. However, nitrogen fertilization is highly dependent on soil nitrogen content and organic matter. The determination of crop N fertilization requirements is related to soil organic matter. However, the authors do not consider this main factor in the manuscript. In such an experiment, soil organic matter (at least) should be analyzed before sowing and after harvest.
----R: Thanks for your strictness review. In the beginning of the study we determine the soil properties of 0–20 cm-deep layer, and the organic matter content were 18.6 g kg–1. But unfortunately we didnot smaple the soil samples at each season’s end. And so ,we cannot give indepth analysis and discussion the effects of soil organic matter.
And base on the review’s helpful suggestion, we will be fully considered the organic matter and nitrogen interaction affected crops growth and productivity in our future studies.
3. The statistical analysis of the data needs to be clarified and corrected. Here, the experimental design is an RCBD with nitrogen fertilization as the only experimental factor and the experiment was conducted for two seasons. I do not understand why each season was analyzed separately. Also, the choice of the post-hoc test must be justified. In such an experiment, Tukey's test is considered the appropriate post hoc test and not the LSD. The ANOVA assumption tests must be reported. The ANOVA output table should be added to the manuscript. There was a significant interaction between season and nitrogen fertilization factors?
----R: Thanks for your strictness review. (1) In this study, the materials were spring wheat and summer maize, so we need to certified the different effects of N application rate on different crops, especially for leaf area index and SPAD, and to confirmed the optimized N application rate for spring wheat and summer maize, respectively, so we should analyzed separately. And we also analyzed the N application rate affected annual grain yield, dry matter accumulation, N accumulation and utilization in an overall perspective conform to the requirements of reviewer. (2) In this study, we think the N application rate was the only factor, and were conducted using a single-factor randomized block design, so take data analysis by LSD to defined the significant or no significant difference level among different N application rate treatments. And we use different lowercase letters to mark the difference. With the review’s helpful suggestion, we take a Tukey's test and found that the result were no different with LSD. (3) Under the review’s helpful suggestion, we take the ANOVA assumption tests, and found if adding ANOVA output tables may makes the results section become cumbersome. Therefore, we believe that the ANOVA output table were no necessary for RCBD data analysis in this MS. (4) The different seasons were the different crops, i.e. spring wheat and summer maize, and so we think analyzed the interaction between season and N application rate didn't fit the statistical requirements.
